# Explainable Machine Learning-Based Risk Prediction Model for In-Hospital Mortality after Continuous Renal Replacement Therapy Initiation

**DOI:** 10.3390/diagnostics12061496

**Published:** 2022-06-19

**Authors:** Pei-Shan Hung, Pei-Ru Lin, Hsin-Hui Hsu, Yi-Chen Huang, Shin-Hwar Wu, Chew-Teng Kor

**Affiliations:** 1Division of Critical Care Internal Medicine, Department of Emergency Medicine and Critical Care, Changhua Christian Hospital, Changhua 500, Taiwan; 181006@cch.org.tw (P.-S.H.); 117610@cch.org.tw (H.-H.H.); 126366@cch.org.tw (S.-H.W.); 2Big Data Center, Changhua Christian Hospital, Changhua 500, Taiwan; 183778@cch.org.tw; 3Department of Nursing, Changhua Christian Hospital, Changhua 500, Taiwan; 73824@cch.org.tw; 4Graduate Institute of Statistics and Information Science, National Changhua University of Education, Changhua 500, Taiwan

**Keywords:** continuous renal replacement therapy, in-hospital mortality, explainable machine learning, SHapley Additive exPlanations (SHAP), local explanation

## Abstract

In this study, we established an explainable and personalized risk prediction model for in-hospital mortality after continuous renal replacement therapy (CRRT) initiation. This retrospective cohort study was conducted at Changhua Christian Hospital (CCH). A total of 2932 consecutive intensive care unit patients receiving CRRT between 1 January 2010, and 30 April 2021, were identified from the CCH Clinical Research Database and were included in this study. The recursive feature elimination method with 10-fold cross-validation was used and repeated five times to select the optimal subset of features for the development of machine learning (ML) models to predict in-hospital mortality after CRRT initiation. An explainable approach based on ML and the SHapley Additive exPlanation (SHAP) and a local explanation method were used to evaluate the risk of in-hospital mortality and help clinicians understand the results of ML models. The extreme gradient boosting and gradient boosting machine models exhibited a higher discrimination ability (area under curve [AUC] = 0.806, 95% CI = 0.770–0.843 and AUC = 0.823, 95% CI = 0.788–0.858, respectively). The SHAP model revealed that the Acute Physiology and Chronic Health Evaluation II score, albumin level, and the timing of CRRT initiation were the most crucial features, followed by age, potassium and creatinine levels, SPO2, mean arterial pressure, international normalized ratio, and vasopressor support use. ML models combined with SHAP and local interpretation can provide the visual interpretation of individual risk predictions, which can help clinicians understand the effect of critical features and make informed decisions for preventing in-hospital deaths.

## 1. Introduction

Acute kidney injury (AKI) is common in critically ill patients in the intensive care unit (ICU). Multinational cross-sectional studies have reported that more than 50% of patients in the ICU have AKI regardless of the severity of AKI [1], and these patients are more likely to have higher mortality and long-term adverse outcomes, including cardiovascular complications, chronic kidney disease, and end-stage renal disease [2]. Continuous renal replacement therapy (CRRT) is the predominant form of renal replacement therapy (RRT), especially for patients with unstable hemodynamic status [3]. CRRT is a blood purification technology that removes excess water and toxins from blood to ensure fluid balance and electrolyte regulation required for organ support [4,5]. Despite advances in CRRT, the prognosis of patients receiving CRRT remains poor, with a mortality rate of 30% to 70% [6,7]. Furthermore, 25% of survivors receiving CRRT require dialysis upon hospital discharge [8]. Therefore, a tool for accurate risk prediction should be developed to determine the outcome of CRRT because a prediction tool provides valuable information that can help physicians make more effective clinical judgments and formulate appropriate treatment strategies as well as help family members make medical decisions.

The outcomes of CRRT depend on various clinical conditions, including patients’ underlying diseases, the severity and duration of renal impairment, renal function at baseline, and the presence of oliguria at the beginning of RRT [3,8,9,10]. Disease severity scores are commonly calculated using various tools, such as the Sequential Organ Failure Assessment (SOFA), Acute Physiology and Chronic Health Evaluation (APACHE) II, and Simplified Acute Physiology Score (SAPS), and are used to predict mortality in critically ill patients; however, these scores have limitations in patients with CRRT [11,12]. With advances in computational capacity in recent years, machine learning (ML) with the modeling of complex mathematical functions for many clinical variables has been used for accurate disease diagnosis and prognosis prediction. Two studies have used ML combined with electronic medical records (EMRs) to predict the mortality and RRT-free survival of critically ill patients receiving CRRT [7,13]. These models exhibited a higher prediction ability than did conventional disease severity scores. However, prediction accuracy of approximately 70% is not sufficient for clinical implementation, and interpretable risk prediction models are not yet available. Although an ML algorithm can indicate the probability of the prognosis of a disease, the algorithm cannot help physicians explain the logic of the decision to family members. Black-box artificial intelligence solutions in medicine without transparency and interpretability are still barriers [14]. To the best of our knowledge, no study has developed and examined an ML-based risk prediction model for CRRT.

Family members are concerned about whether the impaired renal function of patients with AKI admitted to the ICU can be restored or whether other adverse outcomes would occur after CRRT initiation. Even if patients are not optimistic, making the decision to discontinue CRRT is considerably challenging for patients and family members. Family members and physicians aim to provide the best possible care to prolong patients’ survival. When life-sustaining therapies cannot meet treatment goals, many clinical decisions, including those related to palliative care, should be discussed by physicians, patients, and families. The development of personalized and interpretable prediction models is crucial for identifying clinically relevant information hidden in large amounts of data and for creating a notification system that supports the work of physicians. In this study, we established an explainable and personalized risk prediction model for in-hospital mortality after CRRT initiation. This is the first study to use an ML algorithm combined with SHapley Additive Explanations (SHAP) and a local explanation method to provide clinical information to help physicians make better clinical judgments and customize treatment strategies.

## 2. Materials and Methods

### 2.1. Study Participants

This retrospective cohort study was conducted at Changhua Christian Hospital (CCH) in central Taiwan, which is a tertiary medical center with 130 ICU beds in five wards. We identified 3303 consecutive ICU patients who received CRRT between 1 January 2010, and 30 April 2021, from the CCH Clinical Research Database (CCHRD), which contains all electronic medical records including daily CRRT records; data on inpatient care, prescriptions, and clinical visits; laboratory results; and death records. AKI was diagnosed on the basis of Kidney Disease Improving Global Outcomes (KDIGO) criteria for serum creatinine elevation, which was determined by comparing baseline serum creatinine levels at admission with those before CRRT.

We excluded patients with preexisting end-stage renal disease receiving CRRT (*n* = 283), those aged <20 years (*n* = 15), and those with missing laboratory data (*n* = 73). Finally, 2932 eligible patients were included in this analysis (Figure 1). The data of the patients were randomly divided into the training set (80%, *n* = 2345) for developing ML models and the test set (20%, *n* = 587) for examining the performance of each model. The Institutional Review Board of CCH waived the requirement of informed consent and approved the study (IRB No: 210509). The CCHRD contains deidentified data. The researchers of this paper followed the Computer-Processed Personal Data Protection Law and privacy regulations in Taiwan.

### 2.2. Outcomes

The primary endpoint was in-hospital mortality after CRRT initiation that was determined by reviewing death records. The secondary endpoint was mortality at 28 and 90 days after CRRT initiation. Mortality risk was determined in the patients receiving CRRT, and the data of the patients were censored at the end of the respective follow-up periods (at discharge or at 28 and 90 days of follow-up) or at the last date for which data were available.

### 2.3. Feature Engineering

In our study, all features were performed according to the diagnostic criteria for AKI and risk factors associated with ICU mortality [6,7,13,15]. Data were selected based on the data available in our clinical research database, electronic medical records, and our areas of expertise. We collected information on 113 clinical features before CRRT initiation from the CCHRD, including demographics (age, sex, and body mass index [BMI]), APACHE II scores at admission to the ICU, diagnostic criteria of AKI, the timing of CRRT initiation, major diagnostic categories (MDCs), vital signs within 24 h after CRRT initiation, fluid balance, medication use, multiple organ support, serum biochemical data before CRRT, and comorbidities. Laboratory data were used as features for predicting CRRT mortality in the data mining phase, including complete blood count, blood gas analysis, white blood cell (WBC) and differential counts, and all biochemical data such as renal function, lipid profile, nutrition-related tests, etc. We excluded 24 features with a prevalence of <5% from the analysis to ensure variability and improve prediction precision [16]. To select the most favorable subset of features for predicting hospital mortality, we used the recursive feature elimination method with 10-fold cross-validation, repeated it five times, and finally selected 29 features. Table 1 presents the statistical descriptions of the selected features between the training and test sets.

### 2.4. Statistical Analysis and ML Algorithm

Categorical and continuous variables are expressed as numbers (proportions) and medians and interquartile ranges, respectively. The chi-square test was used to compare categorical variables, and the Mann–Whitney U test was used to compare continuous variables.

Figure 2 presents the study framework used for developing the prediction model for in-hospital mortality after CRRT initiation, which included data preprocessing, feature engineering, ML models, and models in global and local interpretation. Four ML algorithms, namely support vector machine (SVM), random forest (RF), gradient boosting machine (GBM), and extreme gradient boosting (XGB) [17], were used to develop a model for predicting in-hospital mortality after CRRT. To select the most satisfactory combination of model hyperparameters, the exhaustive grid search algorithm was used as a hyperparameter tuning tool, and a 5-fold cross-validation procedure was used for the training set. The most favorable hyperparameter for each ML model was determined on the basis of the highest area under the receiver operating characteristic curve (AUROC). ML models were developed using the one-hot encoding of categories and the normalization of continuous features. The Youden index was used to determine the optimal threshold to classify in-hospital deaths that occurred after CRRT initiation. Six evaluation metrics, namely sensitivity, specificity, positive predictive value (PPV), negative predictive value (NPV), F1 score, and accuracy of the test set, were used to compare the prediction ability of each model. Because the study aim was predicting the occurrence of in-hospital mortality after CRRT initiation, we mainly focused on achieving a higher prediction accuracy and F1 score.

A raincloud plot including individual data points (horizontally jittered), density distributions, and statistical inferences of box plots with medians and interquartile ranges was created; this approach enabled the visualization of the distribution of in-hospital mortality after CRRT initiation and predicted scores obtained using ML models. We evaluated the agreement between the predicted probability and observed in-hospital mortality after CRRT initiation by using calibration belts. Kaplan–Meier curves were used to predict the survival rate at 28 days and 90 days.

SHAP is an explainable artificial intelligence technique that helps clinicians understand the results of ML models. Two advantages of SHAP increases the transparency of the model in providing global and local interpretability. Global interpretability lists the most important features in descending order, with the top-ranked features contributing more to the predictive model and having high predictive power. The SHAP value plot can further show the positive and negative relationship between the feature and the outcome variable. Local interpretability provides feature contributions to the model prediction output for an individual patient, which reveal the impact of input features on individual predictions. For clinical practice applications, local interpretability enables physicians to understand key features that affect a patient’s condition and accordingly implement treatment strategies to save the lives of patients receiving CRRT. In this study, all statistical analyses were performed using SPSS, and the ML model was established using R software (version 3.6.2; The Comprehensive R Archive Network: http://cran.r-project.org, accessed on 12 December 2019). All two-sided *p* values of <.05 indicated statistical significance.

## 3. Results

### 3.1. Study Population Characteristics

This study included a total of 2932 patients receiving CRRT, of whom 2024 (60.03%) died in the hospital. The patients who died in the hospital were older and had lower BMI, higher APACHE-II scores, a higher frequency of receiving CRRT > 24 h after AKI, more unstable vital signs (lower blood pressure and SPO2 and higher respiratory rate), and more fluid imbalance before CRRT initiation. Moreover, the patients who died in the hospital were more likely to require multiple organ support before CRRT initiation (including invasive mechanical ventilation, vasopressor support, and vasopressin) and medications (corticosteroids, parenteral nutrition, antibiotics, and furosemide). Moreover, these patients had higher serum potassium, lactate, sodium, and magnesium levels; red blood cell distribution width (RDW); international normalized ratio (INR); and activated partial thromboplastin time (APTT). However, these patients had lower serum creatinine and albumin levels, platelet counts, pH values, and O2 saturation. The prevalence of in-hospital death and the distributions of the features were similar between the training and test sets, except for RDW and APTT. Table 1 lists the features used to develop the ML model.

### 3.2. Model Prediction of In-Hospital Death after CRRT Initiation

As presented in Table 2, the SVM model exhibited a moderate discrimination ability (AUC = 0.750, 95% CI = 0.708–0.792 for the radial kernel; AUC = 0.756, 95% CI = 0.745–0.821 for the sigmoid kernel; and AUC = 0.784, 95% CI = 0.745–0.822 for the polynomial kernel). The RF, XGB, and GBM models exhibited a higher discrimination ability (AUC = 0.816, 95% CI = 0.781–0.851; AUC = 0.806, 95% CI = 0.770–0.843; and AUC = 0.823, 95% CI = 0.788–0.858, respectively). Among the 3 ML models, the RF model exhibited the highest sensitivity (74.69%); however, its specificity and PPV for predicting in-hospital death (75% and 86.38%, respectively), F1 score (80.11%), and accuracy (74.79%) were lower than those of the GBM and XGB models. By contrast, the XGB and GBM models more accurately predicted in-hospital death, with sensitivity of 73.43% and 74.19%, specificity of 80.32% and 78.72%, and PPV of 88.79% and 88.10%, respectively; their F1 scores (80.38% and 80.55%, respectively) and prediction accuracy were the highest (75.64% for both, respectively). Figure 3 presents the discrimination performance of each of the six ML models, as represented by the receiver operating characteristic curve.

The raincloud plot depicted in Figure 4 summarizes the distribution of the predicted scores for the in-hospital death and non–in-hospital death groups. Significant differences were noted in the predicted scores between the groups in the six ML models (all *p* < 0.001, Kolmogorov–Smirnov test), and the median score of in-hospital death was higher than that of non-hospital death (all *p* < 0.001, Wilcoxon rank-sum test). Furthermore, the XGB model overlapped on the 45° dotted line, indicating favorable agreement between predicted probabilities in ML and the actual results determined using the calibration plots (*p* = 0.122 for the XGB model; Figure 5). The Kaplan–Meier curves showed poor survival at 28 and 90 days in the high-risk group (Figure 6).

### 3.3. Model Explanations

Figure 7 presents the plot of the features vital for the XGB model in order of importance according to average absolute SHAP values, which were useful for determining the contribution of each feature to individual predictions. The APACHE II score, albumin level, and timing of CRRT initiation were the top three main features, with SHAP values of 0.300, 0.297, and 0.276, respectively (Figure 7a), followed by age, potassium levels, SPO2, mean arterial pressure (MAP), INR, creatinine levels, and vasopressor use. Higher APACHE II scores; potassium, magnesium, lactate, and sodium levels; MDC scores; RDW; APTT; fluid balance; respiratory rate; and older age as well as lower SPO2, MAP, BMI, platelet count, creatinine levels, and O2 saturation were positively correlated with in-hospital death (Figure 7b). Multiple organ support use (vasopressor, vasopressin, and invasive mechanical ventilation) and medication use (corticosteroids, parenteral nutrition, antibiotics, and furosemide) were positively associated with in-hospital death. The presence of diabetes mellitus was negatively associated with in-hospital death. A nonlinear relationship was observed between pH and in-hospital mortality. Although the distribution of SHAP values was highly dispersed, the correlation of the features with in-hospital death was still consistent with the domain knowledge of most of the features.

Figure 8 depicts four local explanation graphs of randomly chosen patients. Our proposed model correctly predicted the risk of in-hospital death for patients A and B but incorrectly predicted the risk for patients C and D.

Patient A died in the hospital, and the probability of in-hospital death predicted by the ML model for patient A was 91%. Patient A had lower SPO2 (88.93), higher pH (7.21), CRRT initiated >24 h after AKI, a higher sodium level (158), lower O2 saturation (85.3), a lower creatinine level (0.83), and a higher MDC score (24); these factors were significantly positively associated with in-hospital death (Figure 8a). Patient B did not die in the hospital, and the probability of in-hospital death predicted by the ML model for patient B was 36.54%. Patient B was aged 63 years; did not require parenteral nutrition; and had a higher albumin level (3.8), higher MAP (124.36), CRRT initiated within 24 h after AKI, higher BMI (25.14), higher SPO2 (95.64), and normal laboratory values (RDW; APTT; INR; platelet count; pH; and sodium, magnesium, and potassium levels); these factors were associated with a lower risk of in-hospital death (Figure 8b).

The probabilities of in-hospital death predicted using the ML model for patients C and D were inaccurate. Local explanations are presented in Figure 8c,d. The probabilities of in-hospital death for patients C and D were 48.2% and 74.4%, respectively. Despite inaccurate predictions, local interpretations can help clinicians effectively understand patients’ conditions and address key clinical features early on to modify treatment strategies and save lives. Similar results were obtained for the GBM and RF model (presented in Figure A1, Figure A2, Figure A3 and Figure A4), indicating that both the global and local interpretations of the ML model can help in making personalized care recommendations to prevent in-hospital death.

## 4. Discussion

In this study, we developed and validated an interpretable ML-based model to predict the in-hospital death of ICU patients receiving CRRT. Our findings revealed that the XGB, GBM, and RF models exhibited higher discrimination performance with an AUC of 0.80. The result indicated that ML can improve the prediction of in-hospital mortality in ICU patients receiving CRRT. Among the 3 ML models, the XGB model exhibited the most favorable performance and calibration; therefore, we used the XGB model to develop the interpretable ML-based in-hospital death risk prediction model. The XGB algorithm that combines SHAP and the local interpretation framework provides a visual feature importance score that can help physicians intuitively understand the key features of patients’ condition and accordingly modify treatment strategies and explain the condition to the patients’ families.

CRRT is a primary life-support technique used to maintain hemodynamic stability and aid in renal recovery in ICU patients [18]. However, higher mortality rates, ranging from 40% to 70%, have been reported in patients receiving CRRT [6,7]. Therefore, the main objective of CRRT is to prevent mortality and adverse outcomes. APACHE and SOFA scores, which are calculated using clinical and laboratory parameters, are widely used to estimate the risk of mortality in ICU patients [19,20]. However, the efficacy of these scores has not been adequately evaluated in patients receiving CRRT [21]. The accuracy of the newly developed mortality scoring system for AKI with CRRT in predicting mortality in patients receiving CRRT was only evaluated in a small study [12]. A large comprehensive medical database study from Taiwan validated these scoring systems in ICU patients receiving CRRT. The results indicated that the prediction performance of these scoring systems based on scores on days 1 and 3 of CRRT in terms of AUC ranged from 0.55 to 0.67. The average AUC of the discrimination ability was 0.74 on day 7 after CRRT initiation [21].

The use of ML models for data analysis has recently become popular due to a large amount of data in electronic hospital records. Furthermore, ML models can accurately predict mortality in ICU patients receiving CRRT. For example, Kang and colleagues developed an ML-based model to predict mortality in patients receiving CRRT [7]. They reported that the RF model was the most favorable prediction model with an AUROC of 0.782 and observed that the accuracy of an ML-based model in predicting mortality was higher than that of other scoring models. The same research team used an ML algorithm to predict hypotension after CRRT initiation and reported satisfactory performance of the XGB model, with an AUROC of 0.828 [22]. Pattharanitima et al. used ML and deep learning to build a model for predicting renal replacement therapy-free survival and demonstrated that the long short-term memory model with multilayer perceptron architecture exhibited high discrimination performance with an AUC of 0.70 [13]. Studies have investigated the accuracy of other ML-based mortality prediction models in intensive care settings, including in patients with lactic acidosis [23], mechanically ventilated patients [24], and those with COVID-19 and AKI [25]. These findings emphasize the importance of ML in predicting outcomes in critical care settings, especially for patients receiving CRRT.

In this study, we combined ML-based models with electronic medical record data to retrieve information on various clinical characteristics that affect in-hospital mortality, and we determined that the RF, XGB, and GBM models exhibited the most favorable discriminative ability with AUROCs of 0.816, 0.806, and 0.823, respectively. The accuracy of the ML models used in this study in predicting in-hospital mortality in the patients receiving CRRT is comparable to that reported in previous studies [7,10,22,23,24,25]. In particular, our model exhibited higher discriminative power than did the prediction model developed by Kang et al. [7]. In our study, the AUC of the XGB model was not higher than that of the GBM model; however, the PPV, F1 score and accuracy of the XGB model were comparable to those of the GBM model. Moreover, the XGB model was more well-calibrated than the GBM model. Thus, the GBM and XGB models may be suitable to predict in-hospital mortality in patients receiving CRRT.

Currently, ML models used to predict mortality in patients receiving CRRT still employ the black box methodology, which involves determining the relationship between given data (inputs) and outcomes (outputs) instead of creating rules based on knowledge. This study provides new insights into the stumbling block of black-box logic and an explainable approach to artificial intelligence. SHAP reported by Lundberg and Lee (2016) explains how different features affect the results of ML models [26]. SHAP is based on the theoretical optimal Shapley value of a game, a widely used concept in cooperative game theory. The feature values of data instances act as players in the coalition, and Shapley values are the average marginal contributions of features across all possible coalitions. Artificial intelligence should be applied in clinical practice to help clinicians understand the rationale of the predictions of ML models. In addition, providing detailed information can enable doctors to gain better insights and make informed decisions regarding preventing in-hospital deaths instead of blindly trusting or not trusting predictions. Moreover, this information can help physicians explain patients’ condition to the family to ensure transparent decision-making.

We used SHAP values and local interpretation to achieve the best prediction performance and interpretability. SHAP technology has been applied to many clinical problems, including acute exacerbations of chronic obstructive pulmonary disease, venous thrombosis of osteoarthritis, and coronary artery calcification [27,28,29]. Recent studies have applied explainable machine learning to predict hospital mortality in publicly available databases, such as the eICU and MIMIC-III databases [30,31]. To our knowledge, this is the first study to use SHAP in EMRs data to predict in-hospital mortality in ICU patients receiving CRRT. We graphically demonstrated the interpretability of the complicated XBG and GBM models by plotting individual risk prediction.

In our study, the model revealed that the APACHE II score, albumin level, and the timing of CRRT initiation are the three most crucial global features with the highest SHAP values, followed by age, potassium and creatinine levels, SPO2, MAP, INR, and vasopressor support use. Furthermore, the results revealed that a higher APACHE II score, potassium levels, lactate levels, sodium levels, and respiratory rate; multiple organ support use and medication use; and older age as well as lower SPO2, MAP, platelet count, BMI, and O2 saturation were positively correlated with in-hospital death. The findings regarding the importance of variables are consistent with those of a previous study [7,11,12,32,33,34]. In this study, an inverse relationship was noted between diabetes mellitus and in-hospital mortality. A lower creatinine level was associated with in-hospital mortality, whereas a nonlinear relationship was noted between pH and in-hospital mortality. These findings are in line with those of previous studies. The reason for lower mortality in patients with preexisting DM remains unclear. Patients with diabetes might have improved tolerance to acute hyperglycemic episodes during critical illness [35]. Although an elevated serum creatinine level indicates the severity of AKI, a meta-analysis suggested that patients with a lower serum creatinine level at the beginning of CRRT may have an increased risk of in-hospital death [18]. In surgical ICU, mortality was observed to have a U-shaped relationship with pH, with acidemia and alkalemia being independently associated with high mortality [36]. A U-shaped association of pH was observed in patients with CRRT. Future studies should investigate the effect of biochemical laboratory data, including creatinine and pH levels, on in-hospital mortality in patients receiving CRRT.

This study has some limitations. First, the use of data from a single-center electronic database may limit the generalizability of our model, and our findings cannot be applied directly to other medical institutions. Second, we used retrospective data for model construction and validation in this study; thus, some potential risks were not considered. Additional prospective experiments should be performed to validate the prediction model. Third, all models included static features, including demographics, medication use, and multiorgan support use, obtained from the time of admission to the start of CRRT. Time-series data, such as laboratory values, vital signs, and urine output, from ICU admission to CRRT initiation were included in the model as mean values. Future studies should use deep learning including various time-series data for predicting in-hospital mortality. Fourth, the sample size of this study population is moderate; however, the sample size of previous studies on related topics using machine learning to analyze CRRT data is approximately 684 to 2349 [7,13,22], which is smaller than our sample size.

In this study, the XGB and GBM models were used to accurately evaluate the risk of in-hospital mortality in ICU patients receiving CRRT. The model involving the use of SHAP and local interpretation provided visually interpretable individual risk predictions, which may help clinicians understand the effect of key features and make informed decisions regarding preventing in-hospital deaths. In addition, explaining the reasons for the treatment decision to the patient’s family can increase transparency. In conclusion, our model provides objective and interpretable predictions that can help clinicians implement appropriate treatment for patients receiving CRRT based on their specific prognosis.

## Figures and Tables

**Figure 1 diagnostics-12-01496-f001:**
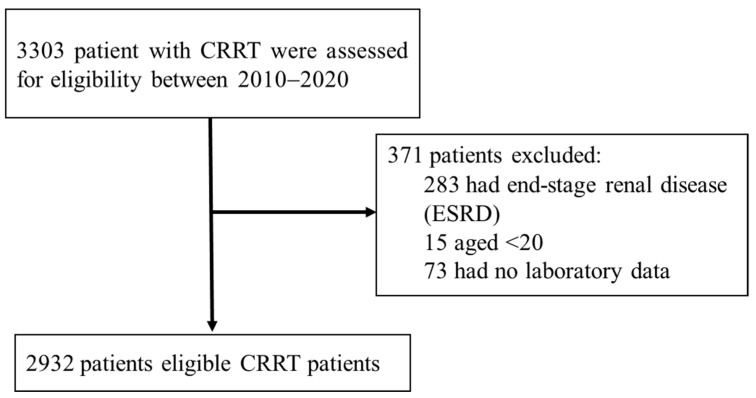
Flow chart of study patients.

**Figure 2 diagnostics-12-01496-f002:**
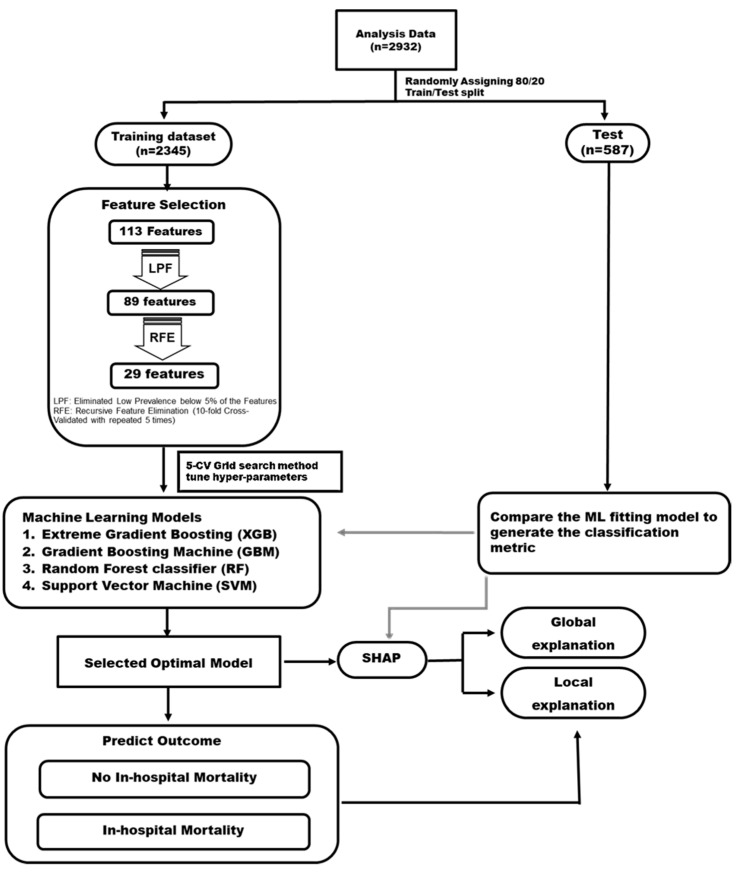
Study framework and feature engineering.

**Figure 3 diagnostics-12-01496-f003:**
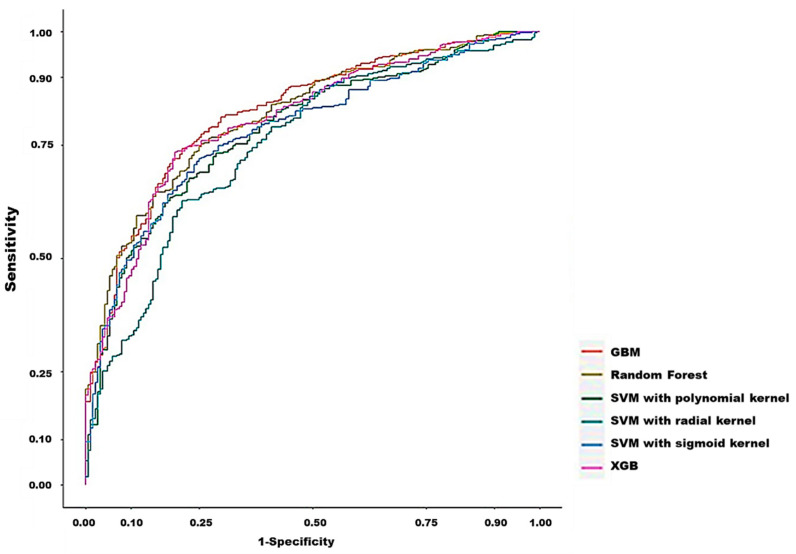
Receiver operation characteristic curves of the models for predicting in-hospital mortality.

**Figure 4 diagnostics-12-01496-f004:**
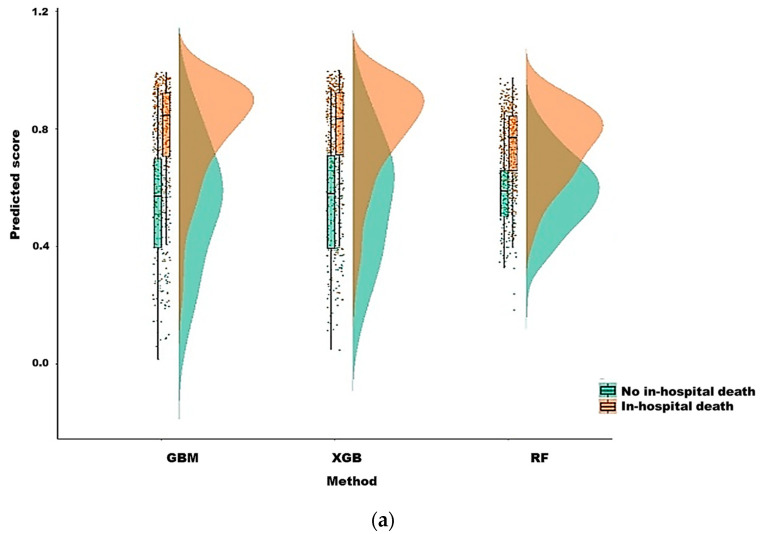
The raincloud plot of in-hospital mortality predicted score in machine learning methods. (**a**) GBM, XGB and random forest, (**b**) SVM with radial kernel, SVM with polynomial kernel, and SVM with sigmoid kernel.

**Figure 5 diagnostics-12-01496-f005:**
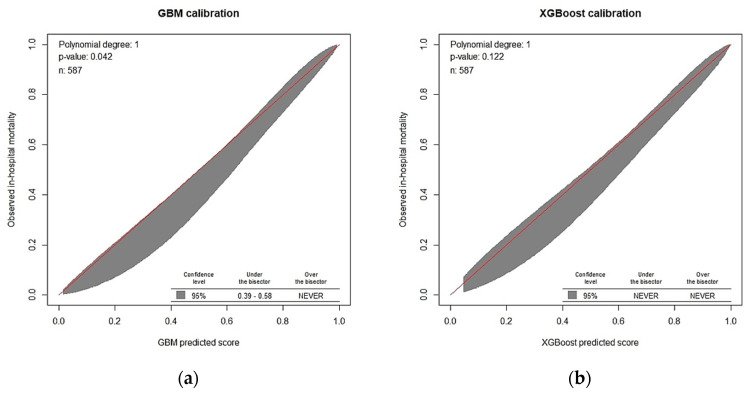
Calibration of machine learning models for predicting in-hospital mortality with calibration belts. (**a**) GBM, (**b**) XGB, (**c**) random forest, (**d**) SVM with radial kernel, (**e**) SVM with polynomial kernel, and (**f**) SVM with sigmoid kernel.

**Figure 6 diagnostics-12-01496-f006:**
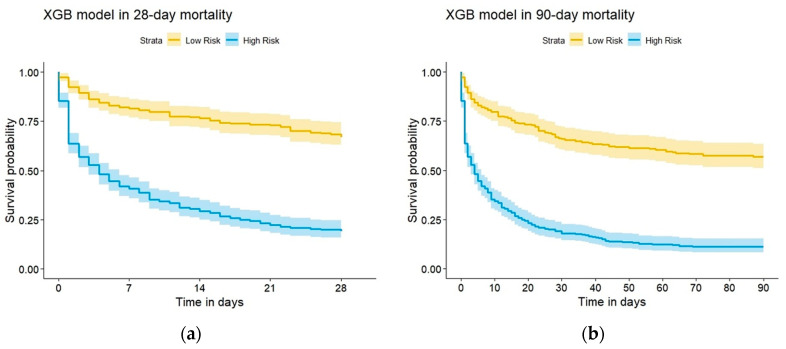
The Kaplan–Meier curve for predicting the 28-day and 90-day mortality. (**a**) 28-day mortality, (**b**) 90-day mortality.

**Figure 7 diagnostics-12-01496-f007:**
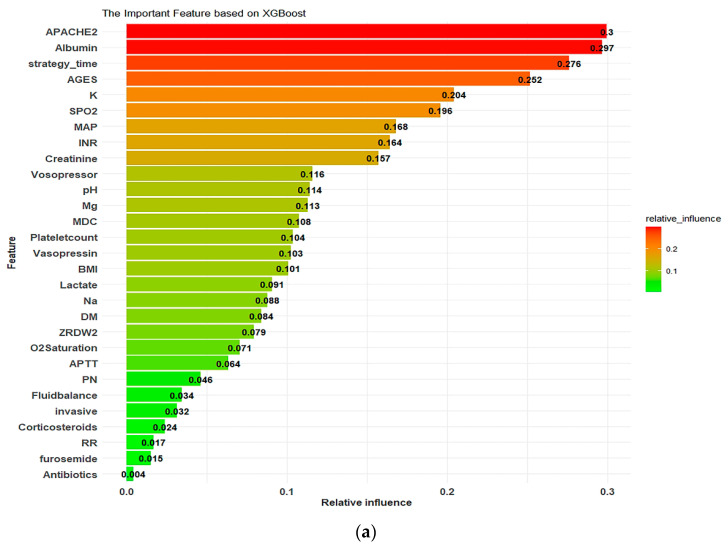
Summary SHapley Additive exPlanations (SHAP) plot. (**a**) Global feature importance in final XGBoost model output. (**b**) Relationship between features and in-hospital mortality in XGBoost model. Diversion on *x*-axis represents effect on model output, with colors used to represent low (yellow) to high (purple) value of predictors.

**Figure 8 diagnostics-12-01496-f008:**
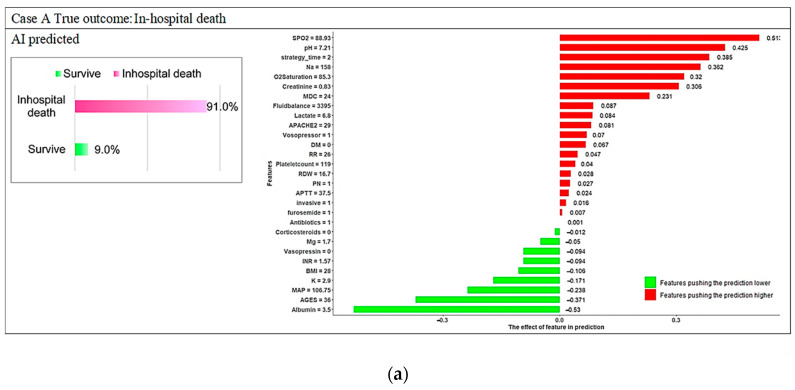
Local explanation plots for individuals with various in-hospital mortality statuses and XGB model predictions. (**a**) In-hospital death and AI predicted in-hospital death; (**b**) non-in-hospital death and AI predicted non-in-hospital death; (**c**) in-hospital death but AI predicted non-in-hospital death; (**d**) non-in-hospital death but AI predicted in-hospital death. Green and red bars correspond to the contribution of the features to the prediction. Green represents a negative value, which decreases the predicted value; red represents a positive value, which increases the predicted value. *X*-axis represents model prediction value; *y*-axis lists the features and their observed values.

**Table 1 diagnostics-12-01496-t001:** Patients’ features in overall cohort and split datasets.

Features	No in-Hospital Death	In-Hospital Death	*p*-Value	Train Dataset	Test Dataset	*p*-Value
Sample size	908	2024		2345	587	
Characteristics						
Age, yr	66 (55–76)	72 (60–81)	<0.001	70 (59–80)	70 (57–81)	0.768
BMI	25.6 (22.5–29)	24 (21.1–27.3)	<0.001	24.5 (21.4–27.9)	24.7 (21.5–28.1)	0.303
MDC	5 (4–8)	5 (4–7)	0.979	5 (4–7)	5 (4–8)	0.198
APACHE II at admission	26 (20–32)	30 (24–37)	<0.001	29 (22–35)	29 (22–35)	0.571
Timing of initiated CRRT						
Early-strategy group ^§^	518 (57%)	962 (47.5%)	<0.001	1200 (51.2%)	280 (47.7%)	0.132
Delayed Strategy	390 (43%)	1062 (52.5%)		1145 (48.8%)	307 (52.3%)	
Vital Sign at CRRT—no. (%)						
Systolic BP (mmHg)	115.7 (105.4–130.3)	108.2 (98.2–119.7)	<0.001	110.6 (100.3–122.8)	109.6 (100.8–123)	0.738
Diastolic BP (mmHg)	60.5 (53.5–70)	57.7 (50.3–65.5)	<0.001	58.6 (51.4–66.8)	58.1 (50.7–66.4)	0.480
Respiratory rate (/min)	18.8 (16.3–21.4)	20 (17–23.3)	<0.001	19.7 (16.7–22.7)	19.7 (16.8–22.8)	0.884
SPO2	97.8 (96–99.1)	96.6 (93.5–98.6)	<0.001	97.1 (94.4–98.8)	96.9 (94.4–98.7)	0.302
Fluid balance before CRRT—ml/24 hr	1663 (500–2992)	2221 (1000–3750)	<0.001	2000 (817–3550)	2000 (835–3429)	0.942
Coexisting conditions—no. (%)						
Diabetes mellitus	368 (40.5%)	642 (31.7%)	<0.001	812 (34.6%)	198 (33.7%)	0.683
Multiple organ support before CRRT—no. (%)						
Invasive mechanical ventilation	719 (79.2%)	1832 (90.5%)	<0.001	2043 (87.1%)	508 (86.5%)	0.709
Vasopressors support with norepinephrine or epinephrine	632 (69.6%)	1792 (88.5%)	<0.001	1931 (82.3%)	493 (84%)	0.348
Vasopressin	178 (19.6%)	843 (41.7%)	<0.001	801 (34.2%)	220 (37.5%)	0.131
Medication use before CRRT—no. (%)						
Corticosteroids	429 (47.2%)	1247 (61.6%)	<0.001	1347 (57.4%)	329 (56%)	0.542
Parenteral Nutrition	678 (74.7%)	1769 (87.4%)	<0.001	1959 (83.5%)	488 (83.1%)	0.813
Antibiotics	828 (91.2%)	1948 (96.2%)	<0.001	2219 (94.6%)	557 (94.9%)	0.800
Furosemide	438 (48.2%)	1104 (54.5%)	0.002	1218 (51.9%)	324 (55.2%)	0.158
Laboratory data before CRRT						
Serum creatinine (mg/dL)	2.6 (1.4–5.2)	2 (1.2–3.6)	<0.001	2.1 (1.3–4)	2.2 (1.3–4.2)	0.640
Serum potassium (mmol/L)	3.9 (3.4–4.4)	4 (3.4–4.8)	<0.001	3.9 (3.4–4.6)	4 (3.4–4.7)	0.699
Serum albumin (g/dL)	2.6 (2.1–3.1)	2.2 (1.7–2.7)	<0.001	2.3 (1.8–2.8)	2.3 (1.8–2.8)	0.448
Lactate, mmol/L	2.9 (1.4–6.6)	4.8 (2.2–10)	<0.001	4.1 (1.9–8.9)	4 (1.9–9.5)	0.906
Platelet count	126 (75–199)	96 (53–165)	<0.001	106 (59–175)	105 (56–177)	0.930
pH	7.4 (7.3–7.4)	7.3 (7.2–7.4)	<0.001	7.3 (7.2–7.4)	7.3 (7.2–7.4)	0.931
Serum sodium (mmol/L)	138 (134–141)	139 (134–144)	<0.001	138 (134–143)	138 (134–143)	0.269
RDW	15.4 (14.3–17)	16.1 (14.8–18.3)	<0.001	15.8 (14.6–17.8)	16.1 (14.6–18.6)	0.039
Mg	2 (1.8–2.3)	2.1 (1.8–2.4)	<0.001	2.1 (1.8–2.4)	2.1 (1.8–2.5)	0.090
INR	1.2 (1.1–1.4)	1.3 (1.1–1.7)	<0.001	1.3 (1.1–1.6)	1.3 (1.1–1.7)	0.785
APTT	36.5 (30.5–54.1)	41 (32.8–69.6)	<0.001	39.2 (31.8–61.5)	40.8 (32.4–69.7)	0.027
O2 Saturation	98.8 (96.6–99.8)	98 (95.1–99.5)	<0.001	98.3 (95.6–99.7)	98.3 (95.7–99.6)	0.955
Outcome						
In-hospital mortality	0 (0%)	2024 (100%)	--	1625 (69.0%)	399 (67.0%)	0.535
28 days mortality	2 (0.2%)	1733 (85.6%)	<0.001	1385 (59.1%)	350 (59.6%)	0.804
90 days mortality	20 (2.2%)	1984 (98%)	<0.001	1607 (68.5%)	397 (67.6%)	0.676

Abbreviations: BMI, body mass index; MDC, major diagnostic category; APACHE, Acute Physiology and Chronic Health Evaluation; CRRT, continuous renal replacement therapy; BP, blood pressure; RDW, red blood cell distribution width; INR, international normalized ratio; APTT, activated partial thromboplastin time. ^§^ Early-strategy group was defined by the renal-replacement therapy was initiated within 24 h of documenting failure-stage acute kidney injury.

**Table 2 diagnostics-12-01496-t002:** Comparison of various models’ performance for predicting in-hospital mortality using test data.

Model	AUC	Threshold	Sensitivity	Specificity	PPV	NPV	F1 Score	Accuracy
Support Vector Machine (SVM) with radial kernel	0.7500	0.6749	62.66%	78.72%	86.21%	49.83%	72.57%	67.80%
Support Vector Machine (SVM) with polynomial kernel	0.7836	0.6588	67.67%	77.13%	86.26%	52.92%	75.84%	73.59%
Support Vector Machine (SVM) with Sigmoid kernel	0.7563	0.6897	71.43%	75.53%	86.10%	55.47%	78.08%	72.74%
Random Forest (RF)	0.8161	0.6587	74.69%	75.00%	86.38%	58.26%	80.11%	74.79%
Extreme Gradient Boosting (XGBoost)	0.8064	0.7234	73.43%	80.32%	88.79%	58.75%	80.38%	75.64%
Gradient boosted machines (GBMs)	0.8227	0.7216	74.19%	78.72%	88.10%	58.96%	80.55%	75.64%

Abbreviations: PPV, positive predictive value; NPV, negative predictive value. The sensitivity, specificity, PPV, and NPV were calculated using Youden’s index.

## Data Availability

Not applicable.

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
