# Peer review of "Explainable Machine Learning-Based Risk Prediction Model for In-Hospital Mortality after Continuous Renal Replacement Therapy Initiation"

_diagnostics, 2022, doi:10.3390/diagnostics12061496_

Round 1
Reviewer 1 Report
Dear Authors,
Thank you for your research. It's impressive study. But I have still two minor questions: why did you choose that laboratory data? You processed all data what you have or you have chosed some specific results?
Have you tried to use random forest for classification?
Reviewer 2 Report
I would like to thank the authors very much for this work. The methodology is largely clear, and the manuscript is well-written as well. It is also good to see that the authors are aware of possible limitations. In general, I believe that the study has a good potential. However, there are a few points that should be considered in the next version, please.
(1)
The article should have much more emphasis on the related work. Recent studies that applied explainable ML in the context of mortality prediction should be discussed, for example:
https://doi.org/10.1109/ICHI48887.2020.9374393
https://doi.org/10.1186/s12874-022-01540-w
(2)
Please elaborate further on the difference between the global and local explanations provided by the SHAP framework. I believe this should be helpful since the audience of this journal is quite inter-disciplinary.
(3)
The Xgboost reference should be cited, please.
Chen, T., & Guestrin, C. (2016). Xgboost: A scalable tree boosting system. In Proceedings of the 22nd ACM SIGKDD International Conference on Knowledge Discovery and Data Mining (pp. 785-794).
(4)
Please consider the (relatively) small dataset as part of the limitations.
(5)
Please re-consider the title. I find it too wordy. Also, it should clearly describe the application of explainable models, which is the core of this work.
Round 2
Reviewer 2 Report
Thanks for accommodating the feedback, I have no further comments.